# Therapeutic Role of Recombinant Human Soluble Thrombomodulin for Acute Exacerbation of Idiopathic Pulmonary Fibrosis

**DOI:** 10.3390/medicina55050172

**Published:** 2019-05-20

**Authors:** Takuma Isshiki, Susumu Sakamoto, Sakae Homma

**Affiliations:** Department of Respiratory Medicine, Toho University Graduate School of Medicine, 143-8541 Ota-ku, Omori Nishi 6-11-1, Tokyo, Japan; takuma.isshiki@gmail.com (T.I.); sahomma@med.toho-u.ac.jp (S.H.)

**Keywords:** acute exacerbation of IPF, thrombomodulin, coagulation, inflammation, HMGB-1

## Abstract

Acute exacerbation of idiopathic pulmonary fibrosis (AE-IPF) is an acute respiratory worsening of unidentifiable cause that sometimes develops during the clinical course of IPF. Although the incidence of AE-IPF is not high, prognosis is poor. The pathogenesis of AE-IPF is not well understood; however, evidence suggests that coagulation abnormalities and inflammation are involved. Thrombomodulin is a transmembranous glycoprotein found on the cell surface of vascular endothelial cells. Thrombomodulin combines with thrombin, regulates coagulation/fibrinolysis balance, and has a pivotal role in suppressing excess inflammation through its inhibition of high-mobility group box 1 protein and the complement system. Thus, thrombomodulin might be effective in the treatment of AE-IPF, and we and other groups found that recombinant human soluble thrombomodulin improved survival in patients with AE-IPF. This review summarizes the existing evidence and considers the therapeutic role of thrombomodulin in AE-IPF.

## 1. Introduction

Idiopathic pulmonary fibrosis (IPF) is a representative subtype of idiopathic interstitial pneumonia. IPF is characterized histologically by a usual interstitial pneumonia (UIP) pattern and radiologically by the presence of typical honeycombing in high-resolution computed tomography images [1]. The clinical course of IPF varies, but respiratory dysfunction is usually chronic and progressive. Kondoh et al. were the first to describe an acute respiratory dysfunction of unknown cause during the clinical course of IPF [2,3], which is now widely known as acute exacerbation of IPF (AE-IPF). AE-IPF is radiologically characterized by new ground glass opacity in bilateral lung fields superimposed on a UIP pattern in patients with extremely severe respiratory dysfunction. Although the 1-year incidence of AE-IPF is not high, 5–15% [4,5], prognosis is poor and reported median survival time was 2.2 months. The 3-month survival rate for AE-IPF was reported to be 30–40% [5,6,7]. AE-IPF is a leading cause of death in IPF patients [8,9].

Although the pathophysiology of AE-IPF is poorly understood, acceleration of underlying fibroproliferative disorders might be involved [3,10]. Histopathological analysis showed interstitial edema, hyaline membrane formation, organizing fibrosis, and findings characteristic of diffuse alveolar damage (DAD) superimposed on a UIP pattern [4,11]. Existing evidence suggests that the pathogenesis of AE-IPF is characterized by increased type II alveolar epithelial cell injury and/or proliferation, endothelial injury, coagulation abnormality, and fibrotic deposition [3,12].

The incidence of AE-IPF is relatively low, and the number of clinical trials has thus been small. Therefore, evidence regarding the treatment of AE-IPF is limited. No randomized clinical trials have investigated AE-IPF; thus, international consensus recommendations are based on weak evidence [1]. In general, corticosteroid pulse treatment is administered in clinical practice. We and other groups reported that when administered with corticosteroids, immunosuppressive agents such as cyclosporine A and tacrolimus might be effective for treatment for AE-IPF [13,14,15,16]. However, other studies reported no clear improvement in patients treated with corticosteroid and immunosuppressive agents [17,18]. Polymyxin-B is a polypeptide antibiotic with bacterial activity toward Gram-negative bacteria and binds circulating endotoxin [19]. In addition to reducing levels of circulating endotoxin, Polymyxin-B direct hemoperfusion (PMX-DHP) reduced blood cytokines, activated neutrophils, and prevented endothelial damage caused by reactive oxygen species [20]. Partial pressure arterial oxygen (PaO_2_) and fraction of inspired oxygen (FiO_2_) ratio (PaO_2_/FiO_2_ ratio) is one of the indexes used to classify the degree of respiratory dysfunction. In retrospective studies, PMX-DHP improved PaO_2_/FiO_2_ ratio [21] and 1-year survival after AE-IPF onset [22]. Donahoe et al. proposed another treatment strategy [23]. Plasma exchange and rituximab treatment improved gas exchange in 9 of 11 patients, and 2-month survival rate was comparable to that of a historical control group. However, because these studies were retrospective and had small sample sizes, the effectiveness of these treatments remains unproven. Lung transplantation is another potential therapy for AE-IPF. Recently, post-transplantation survival among IPF patients including AE-IPF was reported [24]. In the study, IPF patients who transplanted lung during AE-IPF had significantly worse short-term and long-term survival compared to stable patients after lung transplantation. Although it might be beneficial in some part of the patients, given the scarcity of available organs for transplantation, it appeared to be not suitable for standard treatment of AE-IPF.

Optimal treatment for AE-IPF has not been defined, and AE-IPF mortality remains high. Thus, new treatment strategies are urgently needed. Thrombomodulin is a transmembranous glycoprotein expressed on the surface of vascular endothelium. It interacts with multiple proteins to block blood coagulation and inhibit inflammation [25]. Recombinant human soluble thrombomodulin (rhTM)—a human soluble form of thrombomodulin that includes the extracellular domains of thrombomodulin—can suppress abnormal coagulation and excess inflammation [26]. rhTM is being used clinically for treatment of patients with disseminated intravascular coagulation [27]. In this review, we discuss the pathogenesis of AE-IPF, thrombomodulin function, and the potential therapeutic role of thrombomodulin against AE-IPF

### 1.1. Coagulation Abnormalities in AE-IPF

The pathological findings of AE-IPF comprise DAD superimposed on the underlying fibrosis of IPF [28]. Oda et al. investigated 52 autopsy lungs from persons with AE-IPF [29]. Histological analysis revealed DAD in 78.8% of the autopsied AE-IPF lungs. Interestingly, alveolar hemorrhage was observed in 28.8% of lungs, and pulmonary thromboembolism was observed in 17.3% of lungs, suggesting the presence of capillary injury and thrombosis in AE-IPF. Collard et al. investigated the plasma profile of coagulant factors in AE-IPF: total protein C, thrombomodulin, and plasminogen activator inhibitor 1 (PAI-1) levels were higher in AE-IPF patients than in those with stable IPF [12]. Likewise, plasma levels of fibrinogen degradation products (FDP), d-dimer, thrombin–antithrombin complex were higher in AE-IPF patients than in those with IPF and pneumonia and those with stable IPF [30]. Interestingly, serum thrombin–antithrombin complex level was significantly lower in survivors than in nonsurvivors on day 7 after AE-IPF onset. In an analysis of bronchoalveolar lavage (BAL), stable IPF patients had significantly higher levels of coagulant factors (such as tissue factor) and fibrinolytic factors (such as plasminogen activator) than did healthy subjects [31]. In addition, d-dimer and thrombomodulin in BAL were significantly higher in AE-IPF patients than in patients with stable IPF [32]. These results suggest that AE-IPF is accompanied by endothelial injury and coagulopathy. Coagulation and fibrinolysis cascade is shown in Figure 1.

Several lines of evidence indicate that anticoagulant therapy improves fibrosis and inflammation in lungs in animal models. Inhalation of anticoagulants, including heparin and PAI-1 inhibitor, suppressed development of bleomycin (BLM)-induced pulmonary fibrosis in mice [33,34]. These findings support the hypothesis that coagulant abnormalities are associated with lung inflammation/fibrosis and might be a potential therapeutic target for AE-IPF. Although warfarin was ineffective against stable IPF [35], administration of low-molecular-weight heparin improved AE-IPF outcomes. Kubo et al. treated IPF patients with corticosteroids, and low-molecular-weight heparin were added for treatment of AE-IPF. Overall mortality from AE-IPF onset was significantly lower in the anticoagulant group than in those treated with corticosteroid alone [36]. Coagulant abnormality is involved with AE-IPF; thus, anticoagulant therapy might be beneficial.

### 1.2. Inflammation by Myeloid Cells and Lymphocytes in AE-IPF

Pathological analysis of transplanted lungs revealed that lungs of IPF patients who developed AE-IPF had very high levels of inflammatory cells, including neutrophils and lymphocytes [37]. Several studies reported increased numbers of neutrophils in BAL and lung tissue of AE-IPF patients [5,38,39,40]; thus, neutrophilic inflammation may be important in AE-IPF pathogenesis. 

Increased proinflammatory cytokines such as IL-6, and IL-8 were observed compared to stable IPF in the analysis of serum cytokine profile of AE-IPF [12,41,42]. Moreover, IL-8 levels in BAL of AE-IPF patients were higher than in the controls. IL-8 production by activated macrophages was found to attract neutrophils and induce alveolar epithelial cell injury and endothelial injury in the lung [43]. Therefore, activated inflammatory macrophages and neutrophils might be critical in AE-IPF.

The profibrotic roles of alternative activated macrophages were reported in animal models of lung fibrosis and IPF [44,45,46,47]. Prasse et al. found that profibrotic macrophages induced fibrosis through chemokine ligand 18 (CCL-18) and that the CCL-18 concentration in BAL was significantly higher in AE-IPF patients than in those with stable IPF [39,45], which suggests that profibrotic macrophages contribute to AE-IPF pathogenesis.

Activated inflammatory macrophages produce proinflammatory cytokines and chemokines such as IL-8 that attract neutrophil recruitment and activation in the lungs of patients with AE-IPF. Lymphocytic activation might also promote inflammation. Moreover, sustained activation of profibrotic macrophages might be important in the development of irreversible fibrosis during the fibrotic phase of lungs with DAD.

### 1.3. High-Mobility Group Box-1 in AE-IPF

High-mobility group box-1 (HMGB-1) is a non-histone nuclear protein that promotes binding of transcription factors and regulates maintenance of nucleosomal structure. HMGB-1 is constitutively present in nearly all cell types and is passively released when cells are injured [48]. Activated monocytes, macrophages, and dendritic cells also produce HMGB-1 [49,50,51]. Serum HMGB-1 concentration was elevated in patients with sepsis [47,48] and acute lung injury [52,53,54]. HMGB-1 released to the circulation might amplify local inflammation by triggering cytokine production from monocytes/macrophage and endothelial cells [55]. Therefore, HMGB-1 is not only a biomarker that reflects inflammation and tissue damage: it can also induce a subsequent inflammatory response and might be a potential target of inflammatory disorders. The reported endothelial cell receptors of HMGB-1 are receptors for advanced glycation end products (RAGE) [56] and toll-like receptors (TLRs) [57,58]. HMGB-1 promotes chemotaxis and cytokine production involving activation of nuclear factor-κβ via these receptors [59,60]. Acute inflammatory response caused by HMGB-1 leads to neutrophil accumulation in the interstitial and intra-alveolar areas and promotes production of proinflammatory cytokines in the lungs [52,60]. 

Previous studies reported that HMGB-1 mediates coagulation activity. HMGB-1 binds to RAGE on platelets, which leads to platelet aggregation [61]. HMGB-1/TLR4 signaling promotes platelet activation and thrombosis [62]. Moreover, HMGB-1 up-regulates expression of tissue factor in monocytes and endothelial cells [63,64]. Fluza et al. reported that recombinant human HMGB-1 induced production of procoagulant factors and fibrinolytic factors such as tissue plasminogen activator (tPA) and PAI-1 from endothelial cells [65]. In addition, HMGB-1 induces formation of tPA and PAI-1 complexes [66]. These findings suggest that HMGB-1 contributes to thrombosis.

HMGB-1 promotes tissue repair and regeneration. Through RAGE, it induces proliferation of smooth muscle cells in rats and humans [67,68]. Furthermore, increased HMGB-1 activated transforming growth factor-β1 signaling and promoted fibrosis by activating matrix metalloproteinase 9. These findings suggest that HMGB-1 promotes lung fibrosis [69,70].

A study of serum HMGB-1 levels in IPF patients found no significant difference between stable IPF patients and control subjects [71]; however, serum HMGB-1 level was lower in AE-IPF patients after PMX-DHP treatment [72]. In addition, serum HMGB-1 levels gradually increased from AE-IPF onset to day 7 in AE-IPF nonsurvivors [30]. Therefore, serum HMGB-1 level might be a prognostic marker of AE-IPF. We analyzed serum HMGB-1 concentration in AE-IPF and fibrosing nonspecific interstitial patients and compared those values with levels from stable IPF and healthy controls [73]. Serum HMGB-1 level was significantly higher in AE-IPF than in the stable IPF patients and healthy controls. Interestingly, a decrease in HMGB-1 level from day 0 to 7 was a significant predictor of 3-month survival after AE-IPF onset. Thus, serum HMGB-1 level might be strongly linked with AE-IPF disease activity.

Hamada et al. reported that HMGB-1 was highly expressed in alveolar epithelial cells and inflammatory cells in a murine bleomycin model [71]. Another group found that HMGB-1 level in BAL gradually increased after AE-IPF onset, and immunohistochemical analysis revealed that HMGB-1 was highly expressed in alveolar macrophages and alveolar epithelial cells in autopsied lung specimens of AE-IPF patients [74]. Therefore, damaged alveolar epithelial cells and activated alveolar macrophages are potential cellular sources of HMGB-1 in AE-IPF (Figure 2).

HMGB-1 might have a crucial role in AE-IPF pathogenesis through its proinflammatory, thrombotic, and profibrotic function and may therefore be a biomarker of AE-IPF and a potential target for AE-IPF treatment.

### 1.4. Function of Thrombomodulin and Association with AE-IPF

Thrombomodulin is a transmembranous glycoprotein found on the cell surface of vascular endothelial cells encoded by chromosome 20 in humans. It comprises 557 amino acids organized into 5 domains, including the N-terminal lectin-like domain, 6 epidermal growth factor (EGF)-like domain, a serine and threonine-rich domain, a transmembrane domain, and a cytoplasmic domain [75]. Thrombomodulin is essential in regulating blood coagulation/fibrinolysis homeostasis [76].

Thrombomodulin can form a complex with thrombin, a terminal enzyme of the blood clotting process. The region including the fourth, fifth, and sixth EGF-like domains of thrombomodulin are responsible for thrombin binding [77]. Thus, thrombomodulin directly inhibits clotting activity by binding to thrombin. In addition, Thrombin was reported to be a potent stimulus of inflammatory reaction and to promote lung fibrosis. Thrombin disrupts the endothelial junction and increases tumor necrosis factor-α (TNF-α) production from monocytes [78]. Thrombin stimulation induced TGF-β1 production by alveolar macrophages and it leaded elevation of α-SMA expression in lung fibroblasts. Moreover, thrombin inhibition protected against BLM-induced pulmonary fibrosis in mice [79,80]. These findings indicate that thrombomodulin suppresses the thrombotic, inflammatory, and fibrotic response of thrombin by binding to it.

Another function of the thrombomodulin/thrombin complex is activation of protein C. Activated protein C inhibits coagulation by cleaving circulating protein C and inactivating procoagulant factors such as Va and VIIIa [81,82]. Activated protein C also has antineutrophilic inflammatory effects. Recombinant human activated protein C dampened neutrophil chemotaxis, adhesion, and migration [83,84]. Activated protein C inhibited coagulant factors and the pro-inflammatory effects of thrombin by activating protease-activated receptor 1 and its downstream sphingosine-1 phosphate receptor 1 signaling pathway and suppressing the thrombin/PAR11/sphingosine-1 phosphate receptor-3 pathway [85,86]. Taken together, these findings indicate that the EGF-like domain of thrombomodulin exerts anticoagulant, anti-inflammatory, and antifibrotic action by binding to thrombin and activating protein C.

The N-terminal lectin-like domain of thrombomodulin has potent anti-inflammatory activity [25]. Maruyama and colleagues reported that it specifically binds HMGB-1, thereby suppressing inflammatory signal by inhibiting HMGB-1/RAGE interaction [87,88]. Although binding of thrombomodulin and HMGB-1 is reversible, the thrombin and thrombomodulin complex subsequently degrades HMGB-1 to its inactive form and further down-regulates inflammatory responses, such as TNF-α activity [89]. 

Thrombomodulin might also exert an anti-inflammatory effect by regulating the complement system [90]. The N-terminal lectin-like domain of thrombomodulin suppresses activation of the classical, lectin, and alternative pathways of the complement system [91,92]. Thrombomodulin down-regulates the alternative pathway of complement activation by directly enhancing endogenous complement inhibitors that inactivate C3b. In patients with atypical hemolytic–uremic syndrome, genetic mutation of N-terminal lectin-like domains of thrombomodulin had a diminished capacity to activate inactive complement factors [93]. 

In an analysis of thrombomodulin in IPF patients, serum thrombomodulin levels were significantly higher in AE-IPF patients than in stable IPF patients [12]. Moreover, thrombomodulin levels were high in BAL from AE-IPF patients [31]. Pathological analysis showed that thrombomodulin expression in alveolar capillaries was lower in IPF patients than in normal lungs [94]. Furthermore, thrombomodulin expression was significantly lower in AE-IPF lungs than in stable IPF lungs [73]. These results suggest that thrombomodulin is released from microvascular endothelium into the alveolar space and circulation by AE-IPF–induced endothelial injury. Therefore, levels of the active form of thrombomodulin might be lower in the microvascular environment in AE-IPF. 

### 1.5. rhTM Treatment in Animal Models of Acute Lung Inflammation and Fibrosis

Several recent lines of evidence show the therapeutic effect of thrombomodulin in animal models of lung inflammation and fibrosis. In a mouse model of acute lung injury, mice were intraperitoneally administered rhTM. rhTM treatment prolonged survival and improved lung histological scores. Levels of regulatory T cells and anti-inflammatory cytokines such as IL-10 were higher in the lungs of thrombomodulin-treated mice [95].

Fujiwara et al. reported that rhTM improved lung inflammation and fibrosis by means of an anti-inflammatory and anticoagulant effect in several animal models in vivo. A murine BLM lung injury model and TGF-β1 transgenic mice treated with LPS showed less fibrotic response after treatment with rhTM. rhTM inhibited apoptosis of alveolar epithelial cells induced by BLM stimulation in vitro and reduced apoptotic cells in TGF-β1 transgenic mice. The study authors hypothesized that rhTM suppresses lung inflammation and fibrosis by suppressing coagulation abnormality and inflammation, and by its antiapoptotic action [96].

Kida et al. investigated the role of HMGB-1 and thrombin in the fibrotic process and the inhibiting effect of rhTM against this process in vitro and in a BLM mouse model [97]. HMGB-1 and thrombin stimulation induced TGF-β1 production by alveolar macrophages. In addition, thrombin stimulation induced α-SMA expression in lung fibroblasts. Interestingly, these profibrotic signals were dampened by rhTM treatment. Moreover, rhTM treatment significantly decreased lung inflammation and fibrosis induced by BLM instillation in mice in vivo. These results indicate that thrombomodulin has multiple effects on suppressing excess coagulation, inflammation, and lung fibrosis in vivo and may be a promising treatment for AE-IPF.

### 1.6. Clinical Trials of rhTM Treatment for AE-IPF

Several recent studies in Japan investigated the effectiveness of rhTM for AE-IPF (Table 1). Patients were administered rhTM intravenously at a dosage of 0.06mg/kg/day or 380U/kg for 6 days in combination with corticosteroids. The sample sizes were similar and the study results were similarly encouraging. Tsushima et al. examined 20 AE-IPF patients treated with rhTM and compared the findings to those from 6 historical cases. The 28-day survival rate was higher in those receiving rhTM [30]. A single-arm nonrandomized prospective study of rhTM for patients with AE of idiopathic interstitial pneumonias [98] reported that the 90-day mortality rate was significantly lower in 11 rhTM-treated patients than in 11 historical controls. In addition, rhTM administration was an independent predictor of survival in multivariate analysis. Kataoka et al. compared outcomes for 20 rhTM-treated AE-IPF patients and 20 historical cases of AE-IPF. The 90-day survival rate was 65% for the rhTM group, which was significantly higher than that of the control group 

(30%) [31]. In addition, rhTM treatment was a significant predictor of 3-month survival in multivariate analysis. In a single-arm prospective study of rhTM treatment for AE-IPF, 10 patients were treated with rhTM plus corticosteroid pulse therapy. PaO_2_/FiO_2_ ratio and survival rate after AE-IPF onset were significantly better in the rhTM-treated group than in historical controls [99].

We previously compared outcomes of 16 patients treated with rhTM and 25 patients treated with conventional therapy. The former received rhTM (0.06 mg/kg/day) for 6 days as initial treatment, in combination with corticosteroids. As compared with the control group, the rhTM group had a significantly higher survival rate at 3 months (40% vs 69%, *p* = 0.048). A univariate Cox proportional hazards regression model showed that rhTM treatment was a predictive factor for survival. Regarding adverse events, 1 patient in the rhTM group developed mild bleeding [100]. In an extended study of rhTM in AE-IPF, 45 AE-IPF patients were compared to 35 patients receiving conventional treatment. Survival at 3 months and overall survival were significantly better in the rhTM group, which suggests that rhTM has long-term benefits in the treatment of AE-IPF [101].

Serum HMGB-1 level might reflect AE-IPF disease activity and predict survival. Therefore, change in HMGB-1 level was analyzed in AE-IPF patients treated with rhTM. Hayakawa et al. reported that serum HMGB-1 level did not significantly change from day 0 to day 29 after AE-IPF in 7 patients [99]. We examined serum HMGB-1 levels in 36 AE-IPF patients [73]. In the rhTM-treated group, serum HMGB-1 level significantly decreased from day 0 to day 7 after AE onset. However, serum HMGB-1 level did not change in the control group. These findings indicate that rhTM treatment decreases HMGB-1 levels in peripheral blood and might improve outcomes of AE-IPF patients.

Adverse events related to anti-coagulant effect of rhTM were reported in few patients treated in the previous studies. Mild hemoptysis and hematuria were observed in one patient in our study populations [100,101] and hemoptysis in one patient in another study [31]. These adverse events were all improved and any severe bleeding events did not develop.

The results of these clinical studies suggest that rhTM treatment is beneficial in AE-IPF. However, the sample sizes of the studies were too small to conclusively show the effectiveness of rhTM in AE-IPF. In addition, most were single-center retrospective studies. To confirm the effectiveness of rhTM treatment, a multicenter prospective randomized controlled trial is ongoing in Japan.

## 2. Summary

AE-IPF is an acute respiratory dysfunction characterized by alveolar epithelial cell and endothelial cell injury, inflammation, coagulation abnormality, and fibrotic deposition. rhTM suppresses inflammation mainly by binding to HMGB-1 and it also suppresses excess coagulation by forming a complex with thrombin and by activating protein C. Several recent clinical studies suggested the possibility that rhTM improve the prognosis of AE-IPF when it used in combination with corticosteroids.

## 3. Conclusions

We reviewed the pathogenesis of AE-IPF, the therapeutic roles of thrombomodulin in AE-IPF, and evidence from clinical trials. Thrombomodulin is a promising treatment for AE-IPF because of its multiple anti-inflammatory, anticoagulant, antifibrotic effects. A multicenter prospective study to confirm the effectiveness of rhTM treatment is ongoing.

## Figures and Tables

**Figure 1 medicina-55-00172-f001:**
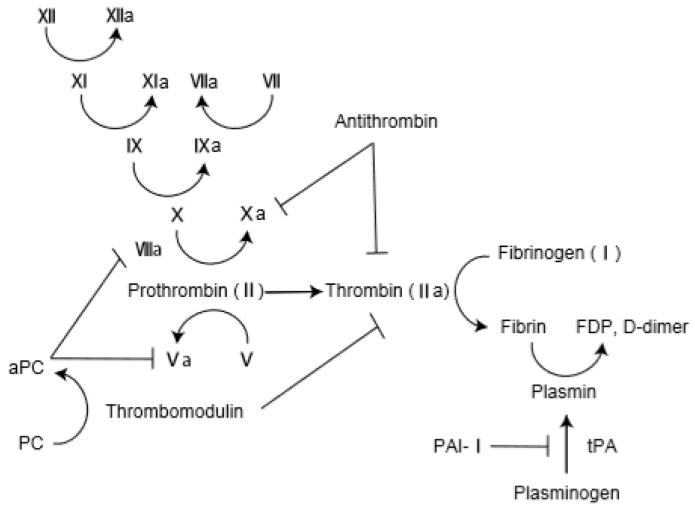
Coagulation and fibrinolysis cascade. PC: protein C; aPC: activated protein C; FDP: Fibrin degradation products; PAI-1: plasminogen activator inhibitor 1; t-PA: tissue plasminogen activator.

**Figure 2 medicina-55-00172-f002:**
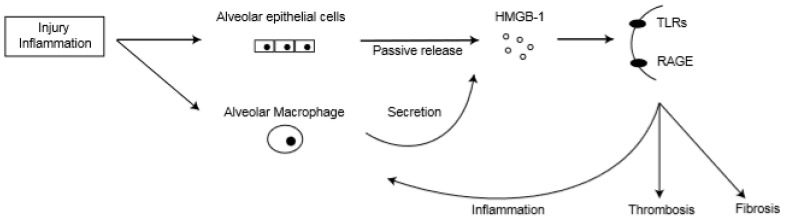
High-mobility group box-1 (HMGB-1) actions in lungs of acute exacerbation of idiopathic pulmonary fibrosis (AE-IPF). TLRs: toll-like receptors; RAGE: receptors for advanced glycation end products.

**Table 1 medicina-55-00172-t001:** Clinical studies of rhTM for acute exacerbation of interstitial pneumonia.

Authors	Background Interstitial Pneumonia	No. of Patients (rhTM vs Controls)	3-Month Survival (rhTM vs Controls)
Abe et al. [98]	IPF and NSIP	11 vs. 11	90 vs. 36%
Kataoka et al. [31]	IPF	20 vs. 20	65 vs. 30%
Hayakawa et al. [99]	IPF	10 vs. 13	60 vs. 15%
Isshiki et al. [100]	IPF	16 vs. 25	69 vs. 40%
Sakamoto et al. [101]	IPF	45 vs. 35	67 vs. 37%

rhTM: recombinant human soluble thrombomodulin.

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
