# Peer review of "Therapeutic Role of Recombinant Human Soluble Thrombomodulin for Acute Exacerbation of Idiopathic Pulmonary Fibrosis"

_medicina, 2019, doi:10.3390/medicina55050172_

Round 1

Reviewer 1 Report

Lines 33-34 - It has been shown recently that the long term outcomes after lung transplant during AE-IPF are significantly worse. Suggest to add this study too: Dotan et al, CHEST 2018 - Effect of Acute Exacerbation of Idiopathic Pulmonary Fibrosis on Lung Transplantation Outcome

Line 79 - BALF is not routinely used. recommend to drop the "fluid" and use BAL (in all the manuscript).

Line 86 - I think the word "inhibitor" is not needed there as it is in "PAI-1".

Line 93 - Suggest to change the wording "hold promise" to "might be beneficial" or something else more "scientific".

Line 102 - Add "the" before "lung".

Line 110 - Change "persons" to "patients".

Line 139 - I suggest to change "indicate" to "suggest".

Lines 207-209 - Has it been shown that levels of Thrombomodulin are lower in the microvascular environment in patients with AE-IPF?

Lines 211-214 - Please describe briefly how were the mice treated? IV? inhalation? else?

Line 220 - Suggest to change "improve" to "suppress".

Line 231 - Please describe briefly the rTM treatment  - route and dose.

Table 1 - please add into the table the study patients disease - IPF? NSIP? else?

I suggest to summarize briefly the findings mentioned in the review in simpler language under "Summary" paragraph, and then give the conclusions in the end. The readers without knowledge of basic science might get lost without a simple summary.

Please add a short paragraph regarding potential side effects of rTM before the conclusions.

Author Response

RESPONSES TO REVIEWERS 

We appreciate the thorough review of our manuscript and feel it has been significantly improved by the revisions. We trust that the article is now suitable for publication in Medicina.

Referee 1.

Lines 33-34 - It has been shown recently that the long term outcomes after lung transplant during AE-IPF are significantly worse. Suggest to add this study too: Dotan et al, CHEST 2018 - Effect of Acute Exacerbation of Idiopathic Pulmonary Fibrosis on Lung Transplantation Outcome

Response: Thank you for the comment. We have cited the relevant study in “ Introduction” section as follows.

“ Lung transplantation is another potential therapy for AE-IPF. Recently, post-transplantation survival among IPF patients including AE-IPF was reported [24]. In the study, IPF patients who transplanted lung during AE-IPF had significantly worse short-term and long-term survival compared to stable patients after lung transplantation. Although it might be beneficial in some part of the patients, given the scarcity of available organs for transplantation, it appeared to be not suitable for standard treatment of AE-IPF. ”

Line 79 - BALF is not routinely used. recommend to drop the "fluid" and use BAL (in all the manuscript).

Response: Wehavechanged the term BALF to BAL in all the manuscript.

Line 86 - I think the word "inhibitor" is not needed there as it is in "PAI-1".

Response: PAI-1 has anti fibrinolysis effect by inhibiting tPA and plasminogen. In the literature (Ref 33), inhibition of PAI-1 prevent the lug fibrosis of BLM model.

Line 93 - Suggest to change the wording "hold promise" to "might be beneficial" or something else more "scientific".

Response: We have changed the word “hold promise” to “might be beneficial”.

Line 102 - Add "the" before "lung".

Response: We have added the “the” before the “lung”.

Line 110 - Change "persons" to "patients".

Response: We have changed “persons” to “patients”.

Line 139 - I suggest to change "indicate" to "suggest".

Response: We have changed “indicate” to “suggest”.

Lines 207-209 - Has it been shown that levels of Thrombomodulin are lower in the microvascular environment in patients with AE-IPF?

Response: Thank you for the comment. Ebina et al showed that (Ref 94) thrombomodulin expression in alveolar capillary endothelial cells was significantly lower in AE-IPF lungs compared to stable IPF and control lungs in their pathological investigation.

Lines 211-214 - Please describe briefly how were the mice treated? IV? inhalation? else?

Response: We have added the following sentence in the manuscript.

“ In a mouse model of acute lung injury, mice were intraperitoneally administered rhTM. ”

Line 220 - Suggest to change "improve" to "suppress".

Response: We have changed “improve” to “suppress”.

Line 231 - Please describe briefly the rTM treatment  - route and dose.

Response: Thank you for the comment. We have added the following sentence in the manuscript.

“ Patients were administered rhTM intravenously at a dosage of 0.06mg/kg/day or 380U/kg for 6 days in combination with corticosteroids. ”

Table 1 - please add into the table the study patients disease - IPF? NSIP? else?

Response: We have added the diagnosis of background interstitial pneumonia in table 1.

I suggest to summarize briefly the findings mentioned in the review in simpler language under "Summary" paragraph, and then give the conclusions in the end. The readers without knowledge of basic science might get lost without a simple summary.

Response: We have added “Summary” paragraph before the “Conclusion”.

Please add a short paragraph regarding potential side effects of rTM before the conclusions.

Response: We added the potential side effect of rhTM in the section “7 clinical trials of rhTM treatment for AE-IPF” as follows.

“Adverse events related to anti-coagulant effect of rhTM were reported in few patients treated in the previous studies. Mild hemoptysis and hematuria were observed in one patients in our study populations [100, 101] and hemoptysis in one patients in another study [31]. These adverse events were all improved and any severe bleeding events did not develop.”

Reviewer 2 Report

This is a nicely written review that argues the case for the potential use of recombinant human thrombomodulin for the treatment of AE-IPF. While I understand that this the focus of this review is on AE-IPF, the authors could consider broadening the review to include more of the existing literature on the coagulation cascade as a target for the treatment of pulmonary fibrosis.

Specific suggested changes

1.     Line 53. The definition of PaO2/FiO2 should be included

2.     Line 102 “and induce alveolar injury” can you please define which cells you are referring to here… epithelial?

3.     Line 172 “and it leaded a-SMA expression” do you mean induced or stimulated a-SMA expression?

4.     The inclusion of a diagram summarising the coagulation cascade and a second figure highlighting the different actions of HMGB1 on lung cells following injury would help the reader.

5.     Sakamoto et al Published a study in 2018 suggesting that overall survival is increased in patients with AE-IPF following treatment with rh TM. Should this study not be included in the Table1 with the others?

Author Response

RESPONSES TO REVIEWERS 

We appreciate the thorough review of our manuscript and feel it has been significantly improved by the revisions. We trust that the article is now suitable for publication in Medicina.

Referee 2.

1.     Line 53. The definition of PaO2/FiO2 should be included

Response: Thank you for the comment. We have added the definition of P/F ratio as follows.

Partial pressure arterial oxygen (PaO2)and fraction of inspired oxygen (FiO2)ratio (PaO2/FiO2ratio) is one of the indexes used to classify the degree of respiratory dysfunction.”

2.     Line 102 “and induce alveolar injury” can you please define which cells you are referring to here… epithelial?

Response: We have corrected the relevant sentence into alveolar epithelial cell injury.

3.     Line 172 “and it leaded a-SMA expression” do you mean induced or stimulated a-SMA expression?

Response: We have corrected the relevant sentence as follows.

 “it leaded elevation of a-SMA expression”

4.     The inclusion of a diagram summarising the coagulation cascade and a second figure highlighting the different actions of HMGB1 on lung cells following injury would help the reader.

Response: Thank you for the comment. We have added the coagulation/fibrinolysis cascade as Figure 1 and HMGB actions as Figure 2.  

5.     Sakamoto et al Published a study in 2018 suggesting that overall survival is increased in patients with AE-IPF following treatment with rhTM. Should this study not be included in the Table1 with the others?

Response: We have added the relevant study in Table 1.